# Reliability and clinical correlations of semi-quantitative lung ultrasound on BLUE points in COVID-19 mechanically ventilated patients: The 'BLUE-LUSS'—A feasibility clinical study

Gábor Orosz[1,2]*, Pál Gyombolai[1], József T. Tóth[1], Marcell Szabó[1,3]

1 Department of Anaesthesiology and Intensive Therapy, Faculty of Medicine, Semmelweis University, Budapest, Hungary, 2 Medical Imaging Centre, Faculty of Medicine, Semmelweis University, Budapest, Hungary, 3 Department of Surgery, Transplantation and Gastroenterology, Faculty of Medicine, Semmelweis University, Budapest, Hungary

* orosz.gabor@med.semmelweis-univ.hu

## Abstract

### Introduction

Bedside lung ultrasound has gained a key role in each segment of the treatment chain during the COVID-19 pandemic. During the diagnostic assessment of the critically ill patients in ICUs, it is highly important to maximize the amount and quality of gathered information while minimizing unnecessary interventions (e.g. moving/rotating the patient). Another major factor is to reduce the risk of infection and the workload of the staff.

### Objectives

To serve these significant issues we constructed a feasibility study, in which we used a single-operator technique without moving the patient, only assessing the easily achievable lung regions at conventional BLUE points. We hypothesized that calculating this 'BLUE lung ultrasound score' (BLUE-LUSS) is a reasonable clinical tool. Furthermore, we used both longitudinal and transverse scans to measure their reliability and assessed the interobserver variability as well.

### Methods

University Intensive Care Unit based, single-center, prospective, observational study was performed on 24 consecutive SARS-CoV2 RT-PCR positive, mechanically ventilated critically ill patients. Altogether 400 loops were recorded, rated and assessed off-line by 4 independent intensive care specialists (each 7+ years of LUS experience).

### Results

Intraclass correlation values indicated good reliability for transversal and longitudinal qLUSS scores, while we detected excellent interrater agreement of both cLUSS calculation

**Data Availability Statement:** All anonymized dataset and image data files are available from the OSF database. DOI: 10.17605/OSF.IO/2V7HA.

**Funding:** The author(s) received no specific funding for this work.

**Competing interests:** The authors have declared that no competing interests exist.

methods. All of our LUS scores correlated inversely and significantly to the P/F values. Best correlation was achieved in the case of longitudinal qLUSS (r = -0.55, p = 0.0119).

## Conclusion

Summarized score of BLUE-LUSS can be an important, easy-to-perform adjunct tool for assessing and quantifying lung pathology in critically ill ventilated patients at bedside, especially for the P/F ratio. The best agreement for the P/F ratio can be achieved with the longitudinal scans. Regarding these findings, assessing BLUE-points can be extended with the BLUE-LUSS for daily routine using both transverse and longitudinal views.

## Introduction

The surges of coronavirus disease 2019 (COVID-19) have a serious impact on the healthcare system worldwide. Thereby, almost every element of the treatment chain—from the prehospital setting to the ICUs—is affected. Several existing and novel LUS (lung ultrasound) protocols were published [1–15], ultrasound- and CT-based studies [1, 4, 16–21] were dealing with the disease-pathognomonic patterns, their types, localization and distribution. Although there is a clear association between CT scan findings and disease severity, in the critically ill, unstable patients it is quite challenging to perform these scans: transporting the patients is relatively unsafe [22] and requires complete staff. Having several patient and staff safety issues taken into consideration, bedside lung ultrasound has gained a key role as a feasible solution for these problems, while providing a diagnostic accuracy which is similar [23, 24] or even superior [25, 26] to conventional radiographic measures.

Unfortunately, it is hard to precisely compare these studies and the results because of the lack of standardization. Recognizing this issue, LUS experts proposed the basics of standardization for the machine settings and scanned regions as well [3, 5, 7, 8, 27].

Besides the machine settings, the optimal type of scanning technique (e.g. longitudinal or transverse) is a matter of discussion which is still controversial [28–31] for daily practice and can be a source of misinterpretation or misdiagnose.

This can lead to confusion for the clinical staff, and may prolong the LUS examination time–thus the contact time with the infectious patient. Furthermore, finding and learning 'COVID-specific' patterns/scoring systems may not be the right way: as some experts have already pointed out, LUS itself is not able to be disease-specific [27, 32, 33].

The widely used and claimed to be 'COVID-19 gold standard' 12- or 14-region LUS protocols [2, 8], however have some drawbacks when applied as part of the daily routine in ICUs: to obtain images from all regions, the potentially unstable, critically ill patients especially those who are ventilated in prone position must be moved or turned in bed; one single operator usually cannot perform it alone in a situation when the infectious risk, availability and overuse of personal protective equipment and workload of the clinical staff is crucial [34]; with the 14-region protocol the dependent dorsal areas are overweighted, constituting a scan-location bias [35].

The above mentioned factors can contribute to the fact that most LUS protocols published to date in COVID-19 pneumonia are time-consuming and cumbersome to perform [15].

Using the well-published, proven scoring systems should be more applicable and reduces the chance for further confusion in nomenclature.

BLUE (or, for precise nomenclature in the ARDS (acute respiratory distress syndrome) patients by the author, Daniel Lichtenstein: PINK [36] protocol is one of the widely used LUS systems in the last two decades [37]. Its popularity is based on its simplicity and well-powered scientific background, however for those who would like to quantify their findings, it may seem to be 'insufficient' and hard to compare for a daily reassessment, furthermore ARDS patients have non-frank profiles, using the BLUE-protocol itself.

*Taking into consideration all the above detailed factors, we hypothesized that a BLUE point-based, single-operator, easy-to-perform methodology for semi-quantitative assessment in the mechanically ventilated COVID-19 patients in the ICU, using the well-known cLUSS (standard coalescence-based lung ultrasound score) and qLUSS (quantitative lung ultrasound score based on % of involved pleura) [29] protocols could be an acceptable way—as we called it: the 'BLUE-LUSS'.*

Our main goal was to prove its feasibility, reliability and clinical value by verifying its possible correlations with disease severity.

## Materials and methods

This is an University ICU-based, single center, prospective, observational study. Between 01 December 2021 and 08 April 2022, a total of 24 patients were enrolled in the investigation, each over 18 years of age, admitted to the ICU with a positive SARS-CoV-2 RT-PCR (reverse-transcriptase polymerase chain reaction) result and respiratory insufficiency requiring mechanical (invasive/non-invasive) ventilation. Exclusion criteria was any living will prohibiting ICU care or declared treatment restriction. Bedside lung ultrasound assessment is a declared part of the daily care at the study site, constructed by the authors of this study and therefore no additional consent was needed. The study was performed with permission of Semmelweis University Regional and Institutional Committee of Science and Research Ethics (Nr. 303/2021 SE-RKEB) and written informed consent was waived. The complete study protocol was conceptualized according to the relevant experiences during the three previous surges of COVID-19. Patients enrolled in this study were treated as per international and institutional standards and guidelines (i.e. mode of ventilation, positioning [supine, prone] of the patient, changing respiratory parameters. . .etc) and study lung ultrasound scans were not used in any way to guide or modify treatment (only exception in obvious emergency situation).

### Patients

The study period coincided with the spread of the 'delta' (B.1.617.2) and the 'omicron' (B.1.1.529) variants in Hungary. We collected relevant clinical information on the patients (e.g. antropometry characteristics, different predictive scores, laboratory findings, oxygenation parameters, vaccination status) as well, detailed later on.

### Ultrasound settings, data collection protocol

We aimed to set up a reproducible, detailed ultrasound protocol. With the following preset of the machine we were able to collect an image pool that was appropriate to precisely observe both the pleura and the artifacts below in sufficient depth to find them without risk of missing important findings [19]. Emphasis was placed upon patient safety as the thermal- (TI) and mechanical indices (MI) were strictly adjusted to the international safety standards [38]. We used a Philips CX50 machine (Philips Healthcare, The Netherlands) with a convex transducer (Philips C5-1 convex probe, 1–5 MHz) and a GE Venue GO machine (GE Healthcare, United

**Table 1. Ultrasound settings.**

| Basic settings | Obligatory value/range |
| --- | --- |
| depth | pleural line + 3–8 cm |
| focus | multifocus: OFF; single focus: at pleural line |
| gain | optimized for main artifacts (A-lines, B-line, consolidation) |
| image-processing features | THI: OFF, XRes: OFF, CrossXBeam: OFF, SRI: OFF |
| MI | < 0.7 |
| TIs | < 1.0 |

Abbreviations: MI: mechanical index; TIs: soft tissue thermal index; THI: tissue harmonic imaging; XRes: speckle noise reduction; CrossXBeam: spatial compounding; SRI: speckle reduction

States of America) with a convex transducer (C1-5-RS convex probe, 1.4–5.7 MHz). Obligatory basic settings are listed in **Table 1**. Based on our clinical and educational experience and in line with international trends in LUS [39], we recorded 4–6 seconds long loops instead of single frames to get the most accurate results possible. The operator was blinded to the clinical course of the patient, not being a member of the treating staff. Loops were recorded with pseudo-anonymization (personal code for the patient) and saved to the hard drive in DICOM (digital imaging and communication in medicine) format, strictly complying with the GDPR (general data protection regulation) rules. Patient data were then transported to an Excel-based dataset with a three-step encryption protocol.

## Imaging protocol

As proposed by Daniel Lichtenstein, the inventor of BLUE/PINK protocol [36, 40, 41], we used the defined points, three per hemithorax, to collect image data, shown in **Fig 1A–1C**. Imaging was performed as a part of the daily routine. Based on previous expert reviews [31] and taking into consideration the heterogenous and 'patchy' characteristic of COVID-19

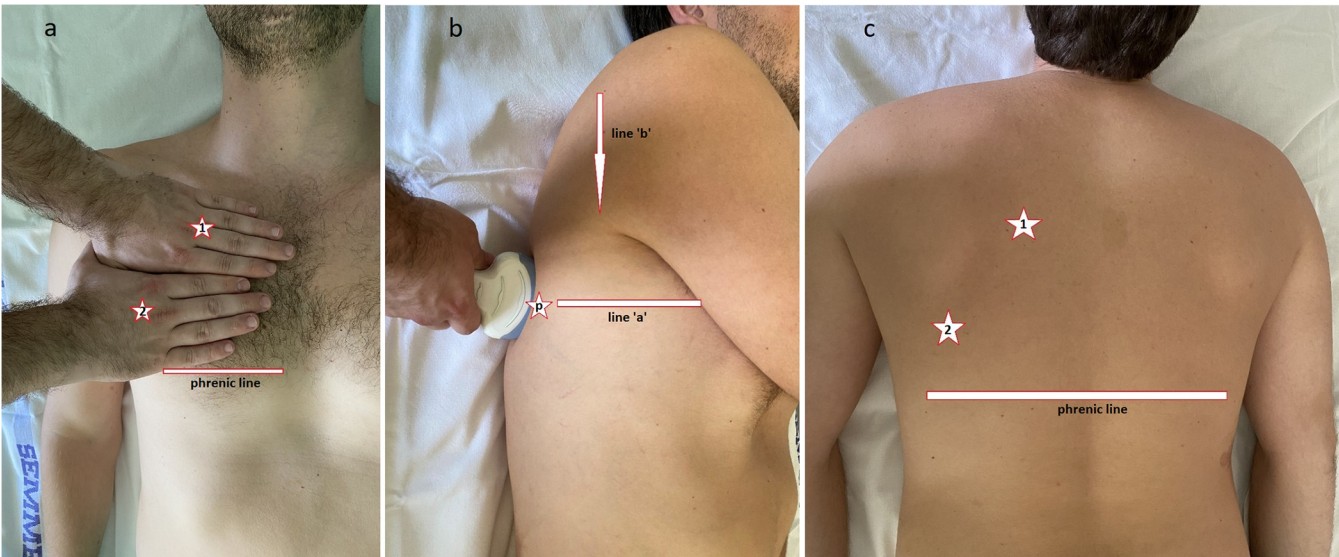

**Fig 1.** a-c. Imaging protocol: BLUE-points. *a: 1 = 'upper' and 2 = 'lower' BLUE-points on supine patients, b: p = 'PLAPS'-point (postero-lateral alveolar and/or pleural syndrome), line 'a' = horizontal line to the 'lower' BLUE-point, line 'b' = posterior axillary line on supine patients, c: 1 = 'upper' and 2 = 'lower' BLUE-points on prone patients.*

pneumonia, we recorded loops both in the longitudinal and transverse directions. The exact locations and directions were tagged with predefined codes. This imaging protocol allowed for assessing the patients in a single-operator manner: no additional help was needed to the operator. As the examinations were all performed by LUS experts with 7+ years of experience, we did not record the exact time length of examination, as it has no additional relevance. Scoring of the loops was performed off-line by default, but in case of readings where an equivalent of emergency was found, the operator obviously reported it to the clinical team as soon as possible. It was the case in one of the examinations when a right-sided acute pneumothorax was accidentally discovered: a life-saving intervention has been performed immediately.

## Scoring protocol

As detailed above, there are several scoring protocols proposed for COVID-19 pneumonia. At the level of loop scoring, we chose two well established, previously validated LUS scoring systems: cLUSS and qLUSS [29], see **Table 2**.

In case of loops where an inhomogeneous LUS pattern was found, we always considered the area with the most serious pathology (i.e. highest cLUSS or qLUSS score),–according to the international standards [2, 29, 42, 43].

Four experts, each with 7+ years of bedside LUS experience, blinded to any of the clinical data related to the patients, scored the loops off-line, one-by-one on a personalized scoring sheet. As data were collected in DICOM format, we used RadiAnt DICOM viewer software (version: 2020.2.3., 64-bit) for loop analysis. There was no time limit or other limitation for loop analysis (e.g. adjusting gain, frame speed. . .etc were accepted).

BLUE-LUSS was calculated on three points per hemithorax with a minimum of 0 and a maximum of 18 points per patient.

Representative images for cLUSS and qLUSS scores, labeled with 3D-Slicer software (origin/version: 4.13.0-2022-01-03 r30524 / ae608a6) [44] are shown in **Fig 2**.

## Statistical analysis

Data were pooled for analysis in Microsoft Excel for Office 365, for the statistical analysis, we used StatsDirect 3.1.20 Statistical Software (Stats Direct Ltd., Grantchester, Cambridge, UK). Continuous patient characteristics are presented as median (interquartile range). Categorical data are shown as the number of cases and percentages. For interrater reliability analysis we used different methods for the analysis of variability observed in the level of single loop scoring and for summarized LUSS at the level of the patients. Single loop characteristics and their LUSS were considered categorical variables and κ values were calculated using *Cohen's* method with either *Fleiss-Cuzick* extension or with *Landis-Koch* extension (two or more than two possible responses, respectively). Intraclass correlations (ICC) for the global LUSS calculations

**Table 2. Scoring protocol: cLUSS and qLUSS.**

| | cLUSS | qLUSS |
|---|---|---|
| **Score 0** | A-lines, MAX two B-lines | A-lines, MAX two B-lines |
| **Score 1** | ≥ three well-spaced B-lines | ≥ three well-spaced B-lines OR coalescent B-lines OR subpleural consolidations occupying ≤ 50% of the pleura |
| **Score 2** | coalescent B-lines | ≥ three well-spaced B-lines OR coalescent B-lines OR subpleural consolidations occupying > 50% of the pleura |
| **Score 3** | tissue-like pattern | tissue-like pattern |

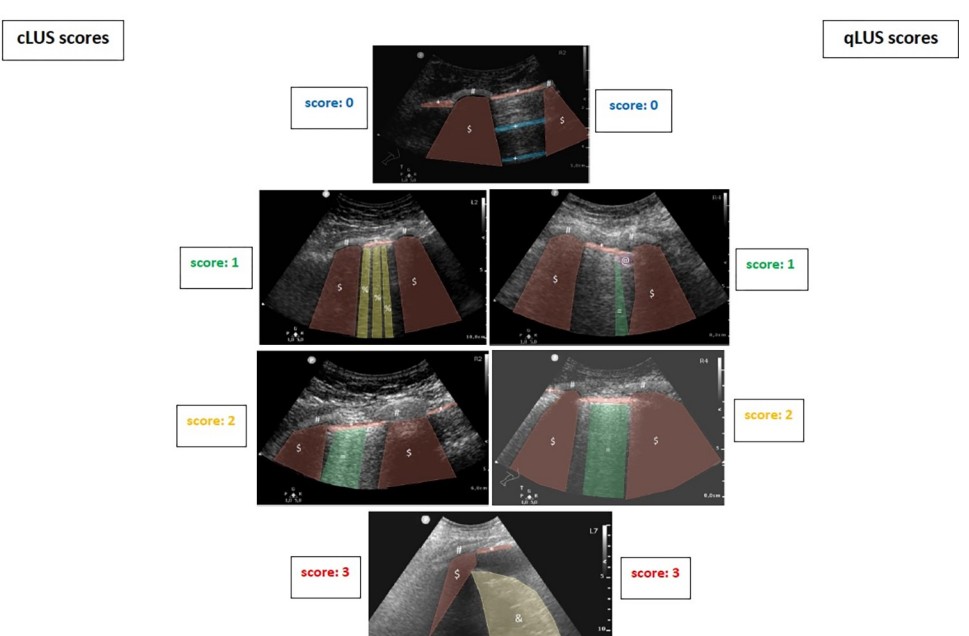

**Fig 2. Representative images of scoring.** Symbols in images $: rib shadow, *: pleural line, #: rib, %: B-lines, @: non-translobar consolidation, =: coalescent B-lines, &: tissue-like pattern (translobar consolidation).

were assessed in a two-way random-effects model [45]. In these analyses we evaluated separately the scores calculated with longitudinal or transverse ultrasound probe orientation. For the evaluation of correlation between continuous clinical variables and LUSS values (median values of the 4 observers) we determined *Pearson's r* coefficients.

Statistical significance was defined as $p < 0.05$. Results of two-tailed tests are presented where applicable. For assuming adequate sample size a type I error of 0.05 and a required power of 0.8 were sought. ICC and Pearson coefficients were the variables of interest, we used an expected reliability of 0.8±0.15 and an r value of at least 0.55 as the basis of the calculations [46]. ICC calculations required the inclusion of 20 patients. Considering that the greatest number of patients was necessary for correlation calculations, the study was powered on the basis of this assumption and we aimed to evaluate 24 patients corresponding to at least 156 ultrasound loops.

## Results

Lung ultrasound scans of 24 patients were eligible for analysis, corresponding to 132 LUS scans. We had to exclude 4 of previously eligible patients because of technical issues: lung scan loops were not able to be totally read from the hard drive after saving or were categorized as damaged by the software and technical support personnel were not able to restore the data structure.

### Population characteristics

Patient demographics and clinical characteristics are described in **Table 3**. Median age was 60 years (46.5–67.5), we observed predominantly male patients (75.0%). CT scans verified at least 50% lung involvement in 50.0% of cases. Majority of the patients were unvaccinated against SARS-CoV-2.

**Table 3. Study population.**

| Patient characteristics N = 20 | |
|---|---|
| Age, years, median (IQR) | 60 (46.5–67.5) |
| Men, N (%) | 15 (75.0%) |
| CORADS class, N (%) | |
| 1 | 1 (5.0%) |
| 2 | 0 |
| 3 | 1 (5.0%) |
| 4 | 3 (15.0%) |
| 5 | 7 (35.0%) |
| 6 | 6 (30.0%) |
| CORADS not available | 2 (10.0%) |
| Patients with defined lung involvement on CT, N (%) | |
| <25% | 3 (15.0%) |
| 25–50% | 5 (25.0%) |
| 50–75% | 7 (35.0%) |
| >75% | 3 (15.0%) |
| CT result not available | 2 (10.0%) |
| Day of LUCI after ICU admission, median (IQR) | 3.5 (3–10) |
| APACHE II, median (IQR) | 11.25 (6–32) |
| CURB-65, median (IQR) | 2 (0–4) |
| WBC at admission, G/l, median (IQR) | 9.4 (6.5–12.5) |
| CRP at admission, mg/l, median (IQR) | 163.2 (69.8–234.5) |
| PCT at admission, ng/ml, median (IQR) | 0.48 (0.26–3.16) |
| Vaccination status, N (%) | |
| none | 14 (70.0%) |
| single | 1 (5.0%) |
| twice | 1 (5.0%) |
| full | 4 (20.0%) |

Abbreviations: CORADS: CoVID-19 Reporting And Data System, IQR: interquartile range, LUCI: Lung Ultrasound scan of the Critically Ill, APACHE II: Acute Physiology And Chronic Health Evaluation II score, CURB-65: Confusion, blood Urea nitrogen, Respiratory rate, Blood pressure age 65 or older pneumonia score, WBC: white blood cell count, CRP: C-reactive protein level, PCT: procalcitonin level.

## Interrater reliability

Calculated κ values from individual loop scoring are presented in **Table 4**.

We observed moderate agreement evaluating individual cLUS and qLUS scores, with a tendency for better reliability of cLUS scores. The highest κ value was detected for the detection of lung sliding, where excellent agreement was verified.

ICC values for summarized cLUS and qLUS scores are shown in **Table 5**.

**Table 4. Interobserver agreement of single loop scores.**

| ultrasound findings/scoring | κ values | CI95% |
|---|---|---|
| cLUSS | 0.56 | 0.52–0.60 |
| qLUSS | 0.48 | 0.44–0.51 |
| lung sliding | 0.92 | 0.78–1.00 |

**Table 5. Interobserver agreement of summarized BLUE-LUS scores.**

| LUS type | ICC value | CI95% |
|---|---|---|
| cLUSS longitudinal | 0.92 | 0.85–0.96 |
| cLUSS parallel | 0.93 | 0.89–0.97 |
| qLUSS longitudinal | 0.79 | 0.64–0.90 |
| qLUSS parallel | 0.87 | 0.77–0.94 |

ICC values indicated good reliability for parallel and longitudinal qLUSS scores, while we detected excellent interrater agreement of both cLUSS calculation methods.

## Correlation of BLUE-LUSS and P/F ratio

Relation of BLUE-LUSS and P/F (partial oxygen tension/fraction of inspired oxygen) ratio is depicted on **Fig 3A–3D**. All of our BLUE-LUS scores correlated inversely and significantly to the P/F values. Best correlation was achieved in the case of longitudinal qLUSS (r = -0.55, p = 0.0129).

## Correlation of LUSS and inflammatory biomarkers

Parallel LUSS (both cLUSS and qLUSS) showed weak to moderate positive correlation with WBC measured on the day of LUCI.

CRP did not correlate to any of the LUSSs, summarized on **Table 6**.

## Discussion

Our first important clinical observation was the quantification of interobserver agreement of the single loop scores. We observed moderate agreement with both cLUSS and qLUSS values, with a tendency for better reliability of cLUSS. Several previous studies dealt with interobserver agreement issues on specific single pathologies (e.g.: B-lines, A-lines, consolidation...etc) and have generally found fair to moderate κ values [39, 47, 48]. However, it is highly important to

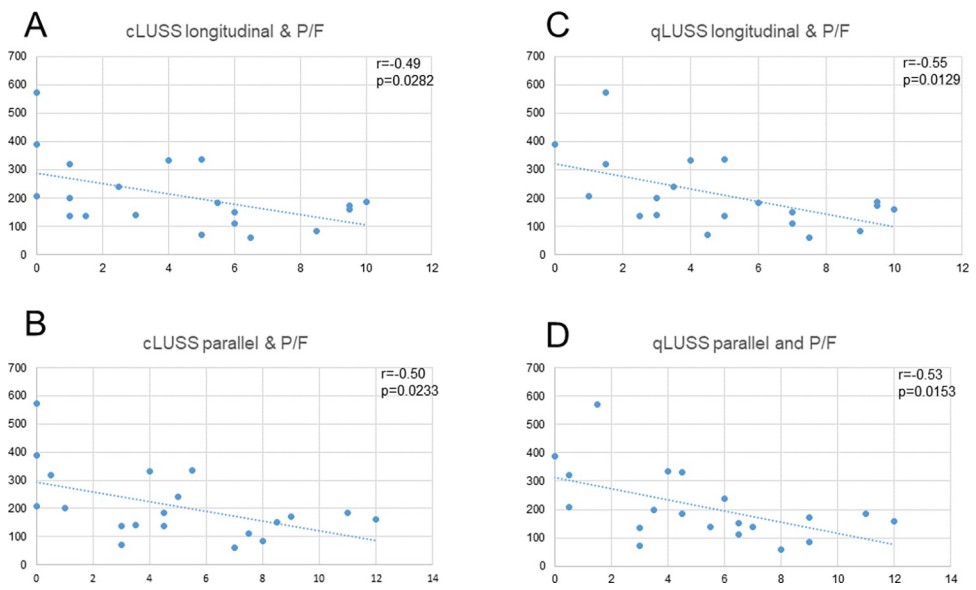

**Fig 3.** a-d. Relation of LUSS and P/F ratio.

**Table 6. Correlation of biomarkers with LUSS.**

| LUSS | biomarker | Pearson's r | p value |
|---|---|---|---|
| cLUSS longitudinal | WBC | 0.41 | 0.0693 |
| | CRP | 0.14 | 0.5602 |
| cLUSS parallel | WBC | 0.45 | 0.0485* |
| | CRP | 0.00 | 0.9903 |
| qLUSS longitudinal | WBC | 0.43 | 0.0591 |
| | CRP | 0.08 | 0.7473 |
| qLUSS parallel | WBC | 0.49 | 0.0266* |
| | CRP | 0.03 | 0.8958 |

verify the agreement not only for single pathologies but also for the calculated single scores. Our results correlate well with the previous findings. The moderate agreement values for single LUS scores (qLUSS and cLUSS) is clinically important and confirms the usefulness of these scoring systems in everyday clinical practice, when the examinations are usually performed by many different observers.

The evaluation of specific single pathologies in previous studies usually did not contain another important issue: the dynamics of the lung. For this reason we also evaluated the single scores for lung sliding, which showed an excellent agreement. As we used only B-mode loops in this study, this finding is in line with the previous observations and recommendations of the inventors of BLUE-protocol, who claim that M-mode is rarely indicated for the assessment of lung sliding, and that practitioners should recognize lung sliding on real-time images first [37].

Although the single loop scores and their coherence are important factors, in day-to-day clinical practice the overall lung condition is the relevant information. Thus, we calculated ICC values for our summarized BLUE-LUSS results, which showed good and excellent correlations for qLUSS and cLUSS methods, respectively. This is a major confirmation for everyday practice and helps to reduce controversies and debates about the reliability of LUCI and issues of comparability [32]. As this study was carried out by experts from the same center, the above findings can be considered to be primarily true in this context. Further investigations are needed to assess the same at other levels of expertise (e.g. novice, trainee, etc) and in different centers, which we plan to perform later on the same dataset.

Good and excellent ICC values may also underline the importance of protocol-based ultrasound settings, data collection and adequate training. The patterns may be easily missed with different device settings, therefore we took into consideration the international proposal of Italian expert colleagues [7, 8, 49], as well as some previous studies comparing CT scans with different LUS findings and artifact types [19]. Based on these it would be useful to update and revise lung ultrasound protocols—also at the level of machine settings. We do not claim that the parameters we used should be chosen, but we would like to draw attention to this issue as it can be an important factor for better correlations and comparability, as previously stated by other experts [27, 28, 33]. Knowing the optimal values of depth, focus and other basic parameters may make LUCI a more powerful tool in the hands of practitioners.

Another important issue is the assessment of the correlation between LUS scores and various clinical parameters and findings. In this respect, a number of possible parameters arise [50–54]. We did not intend to compare our results with CT scans, as these correlations have been extensively studied during the COVID-19 pandemic and are also well documented for the scoring systems we used [1, 4, 16–20]. For similar reasons, mortality was not an end-point for our investigation—the case number was also not planned for this issue. As for laboratory

parameters, we tested the correlation of our BLUE-LUSS results with WBC, PCT and CRP values of the patients from the same day, and did not find any significant correlations. This is not surprising, however, given that the role of ultrasound has already been questionable in this regard. During the pandemic, some studies found correlation with some of the measured laboratory parameters (e.g. CRP and IL-6 (interleukin-6), D-dimer levels) but the results were not consistent [51]. Taking into account all the factors that may influence the above (mainly inflammatory) parameters (e.g. co-infections, comorbidities, latency, plasma half-life etc.), and that they mostly reflect pathophysiological processes from the whole body, it is unlikely that the correlation with lung ultrasound is always present, especially if the assessment is repeated on a daily basis.

The situation is different for parameters that provide information on the patients' actual oxygenation. In our assumption, lung ultrasound should be able to link two very important clinical features: the actual oxygenation status and the current lung aeration of the patient [29, 42, 43, 55]. We were therefore pleased to see that our data showed a significant negative correlation between actual P/F ratio and BLUE-LUSS values of the patients. This linkage is well-described again [12, 54, 56–59], but other scoring systems may have some drawbacks: compared to our BLUE-LUSS approach, they are usually more time-consuming, can be cumbersome to perform, require patient mobilization, all of which can be significant issues in the management of the critically ill. Indeed, although most of the above studies performed lung ultrasound at admission [12, 60–62], a recent study utilized LUCI on a day-by-day basis, and the authors themself declared that the 12-point protocol was very hard to perform, mainly due to staffing and time constraints [63]. This statement by the authors highly underlines the usefulness of our efforts to create a protocol that is easy to implement.

Combining the usage of the conventional BLUE-points with well-powered, 'non COVID-specific' scoring systems can be an optimal way, especially for general everyday use, taking into consideration that in ARDS patients we can observe non-frank profiles. Its important advantages are that it absolutely meets the needs of being fast in time without moving the patient significantly, it is widely known and used (no need to practice a new scanning method) with standardized BLUE-points, and it is usually used on-the-spot in case of sudden changes in the patients' respiratory status on a daily basis.

The scoring systems used here—namely cLUSS and qLUSS—are well-described and widely used, and not thought to be 'specific' for COVID-19 pneumonia. Since cLUSS was the first of the two, maybe practitioners have more experience with it, but qLUSS may have the potential for more detailed discrimination between pathologies, especially inhomogenous or 'patchy' ones, such as COVID-19. In our study we found both of them equally correlated with the P/F ratio, with some tendency for the best correlation in the case of longitudinal qLUSS scans.

Debate on longitudinal vs transversal scanning is still controversial, so in the present study we measured both. Drawback of longitudinal scanning may be that it is thought to be technically more challenging at a non-expert level [31]. On the other hand, the examined lung surface is more extensive—the diagnostic potential can be better [2, 31]. Again, in this study performed by experts we found no significant difference between the two scanning methods, but we plan to perform another study with novices on the same dataset.

Our main goal was to introduce the novel 'BLUE-LUSS' protocol and to prove its feasibility, reliability and clinical value by verifying its possible correlations with disease severity. As this score is not intended to be 'specific' for COVID-19 disease or for any virus variants, it may be used to assess patients with respiratory insufficiency of other origin who are posing a risk for being transported as well. We plan to prove these features of extensive usability in a subsequent study of patients with miscellaneous origin of respiratory insufficiency.

## Strengths and limitations

The main strength of our study is that it is a prospective, feasibility one. We tried to carefully detail all the factors, settings that we used, mainly upon the experiences of previous COVID-19 surges and implement our own clinical experiences, proposing the BLUE-LUSS itself.

Creating a setup which is easy-to-perform, compatible with the everyday workflow, using well-described scoring and scanning protocols and taking into consideration the staff related factors is another important positive issue. Providing some ultrasound setting details for further use may impact the daily routine. It is important to note that training in new protocols and models is a crucial factor for efficiency. With our setup no new skills should be trained, as it utilizes well-practiced, widely known ones, also for the implementation of findings. This may help novice physicians to avoid confusion.

Proving good-to-excellent ICCs and finding important fundamental correlation with the proposed BLUE-LUSS and P/F ratio is notable and can be a possible daily practice in the future as well.

We highlighted the importance of some previously existing score systems and proved their usefulness in this scenario.

The main limitations of our study is its size and that it is single-centered. The sample size was planned to give adequate power for the above detailed results. Image collection and rating of loops in this study were performed by experts in the field of LUCI. Further investigations are needed to validate whether practitioners with less experience, or from different centers can reproduce the results. We plan to set up a study for these issues as well.

## Conclusion

To summarize our results, we found it useful to simplify the semi-quantitative LUCI assessment for everyday clinical practice through some restrictions. Combining the traditional BLUE-points and the well-validated cLUSS/qLUSS scores into a novel 'BLUE-LUSS' value is a feasible option, while extending the transversal scanning planes with longitudinal ones is technically manageable and non-inferior to 'traditional' transversal planes.

BLUE-LUSS can be a general option for LUCI at places of limited or restricted human- and personal protective equipment resources or in case of another surge of COVID-19 pandemic.

Agreeing with large, multicentric studies, we claim that in case of optimal circumstances, enough time and staffing, other scoring systems and scanning protocols can add other important information to the LUCI findings.

## Supporting information

**S1 Table. Correlations of LUSS scores and PCT, APACHE II and CURB 65 pneumonia score.**
(DOCX)

## Acknowledgments

The authors wish to thank the nurse staff of the Semmelweis University Intensive Care Unit for the sustained hard work they did during the pandemic, saving as many lives as possible. The critical reading of the manuscript by Associate Professor Balázs Hauser M.D. PhD (Semmelweis University) is acknowledged with many thanks.

The authors declare no conflicts of interest.

## Author Contributions

**Conceptualization:** Gábor Orosz.

**Data curation:** Gábor Orosz, Pál Gyombolai, József T. Tóth, Marcell Szabó.

**Formal analysis:** Gábor Orosz, Marcell Szabó.

**Investigation:** Gábor Orosz, Pál Gyombolai, József T. Tóth, Marcell Szabó.

**Methodology:** Gábor Orosz, Marcell Szabó.

**Project administration:** Gábor Orosz.

**Supervision:** Gábor Orosz.

**Validation:** Gábor Orosz.

**Visualization:** Gábor Orosz, Marcell Szabó.

**Writing – original draft:** Gábor Orosz, Pál Gyombolai, József T. Tóth, Marcell Szabó.

**Writing – review & editing:** Gábor Orosz, Pál Gyombolai, József T. Tóth, Marcell Szabó.

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
