## [Decision Letter · Decision Letter 0]

6 Sep 2022

PONE-D-22-16527Reliability and clinical correlations of semi-quantitative lung ultrasound on BLUE points in COVID-19 mechanically ventilated patients: The ‘BLUE-LUSS’ - a feasibility clinical studyPLOS ONE

Dear Dr. Orosz,

Thank you for submitting your manuscript to PLOS ONE. After careful consideration, we feel that it has merit but does not fully meet PLOS ONE’s publication criteria as it currently stands. Therefore, we invite you to submit a revised version of the manuscript that addresses the points raised during the review process.

Please revise. 

We look forward to receiving your revised manuscript.

Kind regards,

Academic Editor

PLOS ONE

Journal Requirements:

Reviewers' comments:

Reviewer's Responses to Questions

**Comments to the Author**

1. Is the manuscript technically sound, and do the data support the conclusions?

Reviewer #1: Yes

Reviewer #2: Partly

2. Has the statistical analysis been performed appropriately and rigorously? 

Reviewer #1: Yes

Reviewer #2: Yes

3. Have the authors made all data underlying the findings in their manuscript fully available?

Reviewer #1: Yes

Reviewer #2: No

4. Is the manuscript presented in an intelligible fashion and written in standard English?

Reviewer #1: Yes

Reviewer #2: Yes

5. Review Comments to the Author

Reviewer #1: With interest, I read the manuscript titled: “Reliability and clinical correlations of semi-quantitative lung ultrasound on BLUE points in COVID-19 mechanically ventilated patients: The ‘BLUE-LUSS’ - a feasibility clinical study”.

Lung ultrasound (LUS) is a non-invasive tool for the fast differential diagnosis of pulmonary diseases and is used in different settings in intensive care. LUS pose a significant advantage due to their widespread availability and cost-effectiveness, potentially allowing more patients to access lung imaging, especially during the COVID-19 pandemic. However, the inter- and intraobserver agreement of LUS findings is an issue. This study aimed to assess the interobserver variability of BLUE protocol in LUS.

However, I do have some suggestions about this manuscript. First, one of the main findings of this study is that it is only moderate agreement values for single LUS scores (qLUSS κ value: 0.56 [95% CI, 0.52–0.06] and cLUSS κ value: 0.48 [95% CI, 0.44–0.51]). I suggest the authors further explore the reasons for moderate agreement values for single LUS scores, e.g., different scoring points (0–3) and other lung conditions (consolidation, pneumonia, pulmonary edema, effusion…). Second, it shows good reliability of interobserver agreement in LUS scores after using BLUE points protocol. Could the authors give more discussion about the reasons for interobserver agreement improvement after using BLUE points protocol? Thirds, COVID-19 pneumonia is also one type of severe pneumonia in mechanically ventilated patients. The presentation of COVID-19 pneumonia in LUS should not be different in others of severe pneumonia. Are there any possible reasons why this study focuses on COVID-19 pneumonia patients?

Reviewer #2: This research article from Gábor Orosz and the colleagues evaluate the feasibility of semi-quantitative lung ultrasound on BLUE points, as the authors called the "BLUE-LUSS" protocol, in COVID-19 mechanically ventilated patient by a ICU based, single-center, prospective, observational design. The theme is interesting, yet there are some points need to be clarified.

1. When is the day of LUCI(lung ultrasound of critically ill) ? from ICU admission or from COVID diagnosis was confirmed ?

2. What's the correlation between LUSS and PCT? APACHE II score? and CURB-65 score?

3. The study included patients who required invasive and non-invasive ventilation. How many and what kind of non-invasive ventilation used in the study ?

4. The study period coincided with the spread of both the delta and the omicron variants. Is there any differences in ultrasound finding or BLUE-LUSS score between different variants?

5. The majority of patients have P/F ratio lower than 300 as shown in Figure 3. What procedures (i.e. prone positioning) had done for these hypoxic patients? Did the ultrasound performed before or after these procedures?

6. PLOS authors have the option to publish the peer review history of their article (what does this mean?). If published, this will include your full peer review and any attached files.

Reviewer #1: No

Reviewer #2: No

---

## [Author Response · Author response to Decision Letter 0]

11 Sep 2022

Comments from Reviewer #1 

Comment 1: I suggest the authors further explore the reasons for moderate agreement values for single LUS scores.

Response 1: Thank You for pointing this out. As we highlighted in the ‘Discussion’ of the manuscript (lines 249-257) several previous studies dealt with interobserver agreement issues on specific single pathologies (e.g. B-lines, A-lines…etc) which in general can be much easier for the observer to identify – and have generally found fair to moderate κ values [39, 47, 48]. In this instance, having a moderate agreement value for a much more complex score (i.e. cLUSS and qLUSS), where the observer should identify and analyze several specific single pathologies [see Table 2. Scoring protocol: cLUSS and qLUSS]) can be interpreted as quite a respectable result. Of note, single loops' individual scores and global LUSSs were examined using different statistical methods. Kappa values, widely used for categorical agreement analysis, are more prone to even mild differences. The intraclass correlation coefficient is a commonly applied measure of agreement for continuous data [PMID 29677066]. Generally speaking, it is a more robust statistical method and better describes the patient's global clinical condition. However, as we point out in our manuscript (lines 270-273) this study was carried out by experts. Further investigations are needed to assess the same at other levels of expertise (e.g. novice, trainee). This further study is in progress however.

Comment 2: Could the authors give more discussion about the reasons for interobserver agreement improvement after using BLUE point protocol?

Response 2: Thank You for this remark. As we mentioned above, the single scores can vary very easily, since the several specific single pathologies that we take into consideration can be interpreted slightly differently. As we point out in the ‘Discussion’ section (lines 265-270), the individual loop scores and their coherence are important factors, but in everyday clinical practice the overall lung condition is the relevant information. This is the great value of our scoring system: in real clinical scenarios, it is absolutely realistic for an aggregate score to correlate better with some important clinical (in this case the P/F ratio) value than an individual value – attributable to a single region. However, when an overall score is calculated - summed - for the final BLUE LUS score, these fine differences are somewhat mitigated. 

Comment 3: The presentation of COVID-19 pneumonia in LUS should not be different in others of severe pneumonia. Are there any possible reasons why this study focuses on COVID-19 pneumonia patients?

Response 3: Totally agree. We added some completion in lines 334-339 according to Your remark. However, this remark is the best support for usage of the LUCI in patients with respiratory insufficiency. One of the greatest advantages of lung ultrasound is that it can be used in almost any situation and any disease. However, COVID-19 pneumonia was a new entity: as researchers we felt it our job to validate our previous knowledge in this brand new situation. On the other hand: our BLUE-LUSS, constructed in this situation, is a new approach. We have used the well-known BLUE-points and we also use several - also well-established - scoring systems as discussed in the ‘Discussion’ (lines 314-327) section, but our approach is novel in combining them. We validated this approach on COVID-19 pneumonia patients for simple reasons: a) we were in a critical surge due to the pandemic and we needed some solutions to the problems discussed in ‘Introduction’ (lines 67-78); b) in this hard situation our tertiary intensive care centre was one of the largest that treated critically ill COVID-19 patients in our country. For the future, we plan to ‘generalize’ this new scoring system to other critically ill conditions. 

Comments from Reviewer #2 

Comment 1: When is the day of LUCI? From ICU admission or from COVID diagnosis was confirmed?

Response 1: Thanks for this important question. We intended to define the day of LUCI from the day of ICU admission, and specified it for Your question in Table 3. as well. However, 80% of the patients arrived directly from the ED, and in these cases admission and RT-PCR testing were almost simultaneous (no more than a few hours apart); 20% of the patients arrived through open ward – all of them on the same day when they arrived to the hospital, and so RT-PCR testing was again nearly at the same time (on the same day). To summarize: every patient included in this study was diagnosed with COVID-19 on the day of their admission to ICU. 

Comment 2: What’s the correlation between LUSS and PCT? APACHE II score? And CURB-65 score?

Response 2: Thank you for this remark. As we mention in ‘Discussion’ section (lines 285-300), taking into account all the factors that may influence the above parameters/scores (e.g. co-infections, comorbidities, latency, plasma half-life…etc), and that they mostly reflect pathophysiological processes from the whole body, it is highly unlikely that the correlation with lung ultrasound scores is always present. The role of lung ultrasound is at least questionable in this regard. Other studies have already dealt widely with these issues, and - not surprisingly - the results were not consistent. We provide our results in detail for Your question (S1 Table as for ‘Supporting information’ section) and marked it in the manuscript as well (line 291). These unverified very weak negative correlations cannot be clinically evaluated. In the hypothetical case of similar negative correlations, they may rely on the calibration of these variables to bacterial infections and sepsis, which are later complications of COVID-19 evoking possibly less severe initial oxygenation disorder allowing for less mortality and longer ICU treatment.

Comment 3: How many and what kind of non-invasive ventilation used in the study?

Response 3: Thank You for this request for clarification. We added clarification to the ‘Materials and Methods’ section according to this (lines 104-107). Overall 3 out of 20 of our patients in this dataset were ventilated non-invasively. In this regard, all of these patients were ventilated with positive-pressure ventilation via full face masks. None of these patients were supported with HFNO. We used dedicated NIV-machines (Philips Trilogy EVO; Philips Respironics Amara full face mask). The decisions about mode of ventilation were made as per standard – according to international and institutional guidelines (https://semmelweis.hu/aneszteziologia/files/2020/10/ARDS-es-paciens-leleg-lepesrol-lepesre_nyilv.pdf). 

Comment 4: The study period coincided with the spread of both the delta and the omicron variants. Is there any differences in ultrasound finding or BLUE-LUSS score between different variants?

Response 4: Thanks for this interesting question. Unfortunately, we did not have the opportunity to verify the exact genotypes during the study, so we do not have these data. On the other hand, we believe that lung ultrasound findings are not able to be so specific. Maybe the distribution of the known artefacts can differ, but up to date we have not read any review studies upon this issue. However, one - only peer reviewed – CT study exists which claims that omicron variant was associated with fewer and less severe changes on chest CT compared with the delta variant and patients with omicron variant had a greater frequency of bronchial wall thickening than those with delta variants (1). According to this result – which is a feasible one retrospectively – we would not expect to have marked differences between the LUSS scores. 

Comment 5: What procedures (i.e. prone positioning) had been done for these hypoxic patients? Did the ultrasound perform before or after these procedures?

Response 5: Thank You for this question. We added some explanation and clarification in the ‘Materials and Methods’ section according to this (lines 104-107). In our department where this study was carried out we used the whole spectrum of conventional and non-conventional ventilation modes. According to international guidelines and our experience, we used APRV ventilation mode quite frequently during COVID-19 surges. Our institutional guideline had clear rules about prone positioning, and in our dataset 10% of the patients were ventilated in this position at the time of LUCI. It is important that the trigger for proning was not the LUCI we used for this study but the standard institutional guide (https://semmelweis.hu/aneszteziologia/files/2020/04/Hason-l%C3%A9legeztet%C3%A9s.pdf). So in cases where the patient was in a prone position at the time of LUCI, the examination was performed in prone; when in the time of LUCI the patient was in supine, the examination was performed in supine. VV-ECMO treatment was also available at our university for the treatment of severely hypoxic COVID-19 patients, however none of the patients included in our study was on VV-ECMO at the time of our investigation.

References in this response letter:

1. Yoon SH, Lee JH, Kim B-N. Chest CT Findings in Hospitalized Patients with SARS-CoV-2: Delta versus Omicron Variants. Radiology.0(0):220676.

---

## [Decision Letter · Decision Letter 1]

2 Oct 2022

Reliability and clinical correlations of semi-quantitative lung ultrasound on BLUE points in COVID-19 mechanically ventilated patients: The ‘BLUE-LUSS’ - a feasibility clinical study

PONE-D-22-16527R1

Dear Dr. Orosz,

We’re pleased to inform you that your manuscript has been judged scientifically suitable for publication and will be formally accepted for publication once it meets all outstanding technical requirements.

Kind regards,

Academic Editor

PLOS ONE

Additional Editor Comments (optional):

Reviewers' comments:

Reviewer's Responses to Questions

**Comments to the Author**

1. If the authors have adequately addressed your comments raised in a previous round of review and you feel that this manuscript is now acceptable for publication, you may indicate that here to bypass the “Comments to the Author” section, enter your conflict of interest statement in the “Confidential to Editor” section, and submit your "Accept" recommendation.

Reviewer #1: All comments have been addressed

Reviewer #2: All comments have been addressed

2. Is the manuscript technically sound, and do the data support the conclusions?

Reviewer #1: Yes

Reviewer #2: Yes

3. Has the statistical analysis been performed appropriately and rigorously? 

Reviewer #1: Yes

Reviewer #2: Yes

4. Have the authors made all data underlying the findings in their manuscript fully available?

Reviewer #1: Yes

Reviewer #2: Yes

5. Is the manuscript presented in an intelligible fashion and written in standard English?

Reviewer #1: Yes

Reviewer #2: Yes

6. Review Comments to the Author

Reviewer #1: A relatively brief review in the context of the revised manuscript.

The authors had adequately revised the manuscript regarding the points I mentioned before. I congratulate the authors on completing this interesting manuscript.

Reviewer #2: The authors have addressed all my comments and I have no further question related to this research article.

7. PLOS authors have the option to publish the peer review history of their article (what does this mean?). If published, this will include your full peer review and any attached files.

Reviewer #1: **Yes: **Kuo-Yang Huang

Reviewer #2: No

---

## [Editor Report · Acceptance letter]

6 Oct 2022

PONE-D-22-16527R1 

Reliability and clinical correlations of semi-quantitative lung ultrasound on BLUE points in COVID-19 mechanically ventilated patients: The ‘BLUE-LUSS’ - a feasibility clinical study 

Dear Dr. Orosz:

I'm pleased to inform you that your manuscript has been deemed suitable for publication in PLOS ONE. Congratulations! Your manuscript is now with our production department. 

Kind regards, 

on behalf of

Dr. Robert Jeenchen Chen 

Academic Editor

PLOS ONE